# Using Genomic Selection to Develop Performance-Based Restoration Plant Materials

**DOI:** 10.3390/ijms23158275

**Published:** 2022-07-27

**Authors:** Thomas A. Jones, Thomas A. Monaco, Steven R. Larson, Erik P. Hamerlynck, Jared L. Crain

**Affiliations:** 1USDA-Agricultural Research Service, Forage & Range Research Laboratory, 696 North 1100 East, Logan, UT 84322, USA; tom.monaco@usda.gov (T.A.M.); steve.larson@usda.gov (S.R.L.); 2USDA-Agricultural Research Service, Range & Meadow Forage Management Research Laboratory, 67826-A Highway 205, Burns, OR 97720, USA; erik.hamerlynck@usda.gov; 3Department of Plant Pathology, Kansas State University, 1712 Claflin Road, 4024 Throckmorton PSC, Manhattan, KS 66506, USA; jcrain@ksu.edu

**Keywords:** bluebunch wheatgrass, ecological restoration, genomic selection, native plant, *Pseudoroegneria spicata*

## Abstract

Effective native plant materials are critical to restoring the structure and function of extensively modified ecosystems, such as the sagebrush steppe of North America’s Intermountain West. The reestablishment of native bunchgrasses, e.g., bluebunch wheatgrass (*Pseudoroegneria spicata* [Pursh] À. Löve), is the first step for recovery from invasive species and frequent wildfire and towards greater ecosystem resiliency. Effective native plant material exhibits functional traits that confer ecological fitness, phenotypic plasticity that enables adaptation to the local environment, and genetic variation that facilitates rapid evolution to local conditions, i.e., local adaptation. Here we illustrate a multi-disciplinary approach based on genomic selection to develop plant materials that address environmental issues that constrain local populations in altered ecosystems. Based on DNA sequence, genomic selection allows rapid screening of large numbers of seedlings, even for traits expressed only in more mature plants. Plants are genotyped and phenotyped in a training population to develop a genome model for the desired phenotype. Populations with modified phenotypes can be used to identify plant syndromes and test basic hypotheses regarding relationships of traits to adaptation and to one another. The effectiveness of genomic selection in crop and livestock breeding suggests this approach has tremendous potential for improving restoration outcomes for species such as bluebunch wheatgrass.

## 1. Introduction

Land disturbances, species invasions, novel ecosystems, and climatic change are rapidly increasing worldwide. Because these challenges are both accelerating and difficult to reverse, a compelling need exists to develop plant materials that maintain native ecosystems, conserve natural resources, and deliver ecosystem services valued by humanity. For example, novel native plant materials may be advantageous when they offer the best opportunity for restoration success. This may occur when a restoration site has become altered beyond the local population’s ‘adaptive envelope’ [1], as when ecosystems have moved away from historical norms due to climate change [2,3], modified soils, or changes in vegetation [4,5]. Under such subpar conditions, plant populations that can tolerate augmented multiple stresses become more attractive for restoration relative to local populations that have evolved under different conditions.

Breed et al. [6] recently suggested the use of genomics to delineate plant provenances, to identify genetic adaptations for resilience, and to propagate novel genotypes through gene editing. The purpose of this synthesis paper is to illustrate how the emerging tools of genomic selection can be used to develop more effective native plant materials with key functional traits for ecosystem restoration. Genomic selection, based on recent dramatic advancements in molecular genetics technology, provides a more efficient path to plant material development than has been hitherto available. By employing genomic selection, it is possible to capitalize on genes controlling traits that are of value in target environments [7,8,9,10,11], leading to improvements in population performance.

### 1.1. Technical Advantages of Genomic Selection

Genomic technologies sufficient to scan the entire genome of cattle have been available for at least two decades [12,13], with great potential for many other domesticated crop, horticultural, and livestock species [14,15]. Selection can now be based on known genotypes based on discretely readable DNA sequences rather than on continuous phenotypic variables, which are usually confounded by environmental effects and experimental error. This has the potential for making selection more efficient and for making ecophysiological modeling of genotype–environment interactions more tractable [16]. Furthermore, the experimental unit of selection becomes more precise, moving from the whole-plant level to the gene level. Because diploid individuals display one (homozygous state) or two (heterozygous state) copies of a given allele, many replicates of a gene can be identified from across the population, increasing statistical power and experimental efficiency relative to clonal or seed replication [17]. Moreover, with enough genetic markers, another form of replication based on exact genetic similarities or kinships among individuals can be established, which is more accurate than traditional forms of replication based on pedigree relationships [18,19,20,21,22].

Genomic selection may dramatically improve the efficiency of selection relative to traditional methods. Using modern laboratory techniques and equipment, larger numbers of plants may be evaluated than would be previously possible at a fraction of the cost required for replicated field evaluations. For traits that are costly (e.g., C-isotope ratio), laborious (gas exchange), or inconvenient (persistence) to routinely phenotype, once DNA sequences associated with the trait are determined, the traits need not be remeasured. Seedlings may be screened and selected for traits expressed only in mature plants, reducing the length of time required to complete a cycle of selection. This has been invaluable for selection in long-lived trees [23,24]. However, it is also being applied to intermediate wheatgrass (*Thinopyrum intermedium* [Host] Barkw. & D.R. Dewey), a species being domesticated as a perennial grain crop [25] and to other perennial plants [14,26]. Exposure to multiple cycles of selection under contemporary changed-climate conditions may be advantageous for adaptation, especially for long-lived species that may have last recruited seedlings decades or centuries ago during an infrequent favorable climatic pulse that is no longer expected to occur [27,28].

### 1.2. Potential Advancements in Ecological Genetics

Besides the direct conservation benefits of more effective plant materials, we expect additional benefits for basic science [15,16]. This research can illuminate the intersection of plant ecology, physiology, and genetics through the prism of functional traits by elucidating how genes and genomes contribute to the traits that direct ecosystem processes. Several examples follow:Adaptive traits can be genetically characterized and mapped to chromosome position. This is particularly advantageous for complex traits, i.e., those controlled by multiple loci and subject to environmental influence.Allelic variants, i.e., alternative forms of the same gene, can be identified, allowing selection for alleles and quantification of their population frequencies.A comprehensive genomic-based model may be developed. As more data are accumulated, the model may be improved, thereby reducing experimental error, increasing the precision of trait-mean estimates, and increasing heritability.Trait combinations (plant syndromes) within or among populations can be identified and genetically characterized. Of particular interest are trait combinations that are adaptive under emerging environmental scenarios. Individuals or populations may be assigned to syndrome using multivariate analysis.Development of isogenic trait-contrasting subpopulations is made possible through genomic selection or marker-assisted selection. Such comparisons can determine whether a given trait is favorable, neutral, or unfavorable at various levels of environmental variables and in various genetic backgrounds. This facilitates rigorous testing of hypotheses involving trait function and correlated trait responses.

## 2. Plant Material Performance

A critical need in dry-land revegetation is the development of native plant materials that are better adapted to a multiplicity of sites. They must be better able to establish, reproduce, and persist than those currently available. This may be accomplished by any combination of (1) adaptive genetic variation for functional traits that confer ecological fitness, (2) phenotypic plasticity that confers adaptation to the occupied environment, and (3) the ability to evolve in response to local environmental conditions, i.e., local adaptation.

### 2.1. Adaptive Genetic Variation

Identifying adaptive genetic variation requires testing plant material for performance [29,30,31]. Evaluation of plant material entails concentrated experimental effort. For some traits, greenhouse or transplanted field experiments may suffice for the trait in question. However, to test overall adaptation, seeded field trials are preferred, as seedling emergence, rather than post-emergence mortality, is typically the greatest limiting factor to obtaining an adequate stand [32]. Hereford [33] showed that adaptive trade-offs among environments are not strong enough to prohibit simultaneous adaptation to multiple environments. The desired genotype capitalizes on environmental conditions that favor it without being subject to severe negative trade-offs in alternate environments. Richards et al. [34] have referred to such genotypes as “jack-of-all-trades master-of-some”.

### 2.2. Phenotypic Plasticity

Identifying plant material with general adaptation across environments, conferred by phenotypic plasticity for adaptive traits, may allow species to colonize a diversity of environments [35,36]. When phenotypic plasticity enhances fitness, it may be considered adaptive [34]. High-plasticity species may function as ecological generalists, while adaptation of low-plasticity specialist species may be more limited [37]. While plasticity itself is subject to natural selection [34,35], it may inhibit evolution, as more-plastic entities are less impacted by natural selection [37].

### 2.3. Potential for Local Adaptation

Natural selection for local adaptation may be responsible for the observed ability of species to colonize disparate environments. For example, local adaptation may be present in invasive species that have only been present in their invasive range for relatively short periods. This suggests that local adaptation may be conferred by rapid evolution [38], particularly for self-incompatible species [39]. While much of the research on adaptation and plasticity has been conducted with invasive species, native species appear to follow the same assembly rules as invasive species [40]. Populations may develop de novo local adaptation via rapid evolution by exploiting high genetic variation [39,41,42], i.e., assisted evolution [4]. The phenomenon of rapid evolution for local adaptation may explain how invasive species are able to occupy a wide range of habitats [39]. Genetically augmented populations of native species may be useful for restoration applications [43], an approach termed admixture provenancing [44].

### 2.4. Issues Surrounding Selection

For cross-pollinated species such as bluebunch wheatgrass, genomic selection has the potential to address the elephant in the room, i.e., the inadequate performance of current native plant materials amid worsening environmental conditions [10]. As with any selection program, a potential concern is the possibility of narrowing the genetic base and resultant inbreeding. This involves two related issues. The first issue involves the alleles controlling the traits under selection. These alleles constitute a tiny portion of the entire genome, thus, such selection does not lead to highly inbred populations, though it could lead to a homozygous state at the affected loci. The second issue, that of reduced genetic variance and inbreeding, is a general consequence of either natural selection or artificial selection. Nevertheless, intentional steps may be taken to either minimize or reverse such losses in a selection program. Minimizing losses of genetic variation can be accomplished by maintaining a high effective population size (N_e_), i.e., including many parents, in each cycle of selection [10]. Reversing losses may be accomplished by the introduction of immigrants, even when descended from the same base population [45]. This approach is effective because, while loss of genetic variation increases with intensity and occurrence of selection, it is a random stochastic process. Hence, loss of genetic variation under mild selection is unlikely to be for the same allele between any two populations selected from the same base. This means that an allele that randomly persists in one population may reverse the random loss of the same allele in another population upon introduction of individuals from the former population into the latter population. Finally, loss of genetic variation can be monitored and quantified with neutral genetic markers through the course of selection to determine whether corrective action is necessary.

## 3. Bluebunch Wheatgrass: A Restoration Workhorse Species

Bluebunch wheatgrass (*Pseudoroegneria spicata* [Pursh] À. Löve; BBWG) is a common perennial Triticeae bunchgrass species in sagebrush-steppe plant communities of the Intermountain West, USA (see Appendix A). Its distribution ranges from the eastern side of the Pacific Coastal Mountains to the western Great Plains and from the Yukon Territory to northern Mexico [46]. It is a climax species of wide ecological amplitude that commonly co-occurs with Wyoming big sagebrush (*Artemisia tridentata* Nutt. ssp. *wyomingensis* Beetle & Young) on well-drained medium-textured soils [47,48]. Bluebunch wheatgrass is predominately diploid (2*n* = 14), but autotetraploid (2*n* = 28) populations occur in the northern portion of the species’ distribution [49]. This species is predominately cross-pollinating [50], making it an acceptable candidate for recurrent selection.

Bluebunch wheatgrass is the most widely used bunchgrass for land-rehabilitation seedings in the Intermountain West. As its seed supplies are plentiful and reasonable in cost, it is the best regional example of a native ‘workhorse’ species. Available plant materials include ‘Whitmar’ (released 1946), ‘Goldar’ (1989), P-7 Germplasm (2001), Anatone Germplasm (2004), and Columbia Germplasm (2015). In addition, geographical zones based on genetic similarity and traits that respond to natural selection have been identified for use in seed transfer guidelines [51,52,53].

However, available plant materials are still suboptimal for reliable performance in rangeland seedings, particularly on sites at the low end (<310 mm) of the average annual precipitation range where BBWG is found [30]. Despite acceptable establishment [31], BBWG displays lower productivity [54] and persistence [31] in comparison with introduced Triticeae wheatgrasses. This grass has suffered reduced abundance due to susceptibility to grazing and annual weed invasion [47]. Increased temperatures and drought may also favor downy brome (*Bromus tectorum* L.) as a competitor to BBWG [55]. Genomic selection applied to BBWG populations could be useful for ameliorating these issues that negatively impact its adaptation.

## 4. Bluebunch Wheatgrass: Plant Traits

We identified traits that may be important for any of five demographic stages (germination, emergence, seedling establishment, reproductive output, and persistence), as well as traits relating to developmental physiology (Table 1). We also associated each trait with one or more of six trait categories: (1) growth, (2) abiotic-stress response, (3) competitive-stress response, (4) defoliation-stress response, (5) reproductive output, and (6) persistence. To persist under current disturbance regimes and projected climate scenarios of the future, bunchgrasses must be able to establish quickly under cool temperatures, to consume excess soil resources to limit weed competition, to tolerate low soil moisture and defoliation, and to reproduce by seed and persist under wildland conditions.

### 4.1. Growth

Seed mass and seed density may have strong impacts on seed germination, seedling emergence, and seedling vigor. Across naturally occurring BBWG populations, seed mass has been positively correlated with both seedling emergence from a 4-cm seeding depth and seedling shoot dry-matter [88]. Populations with low seed mass germinated earlier and produced seedlings with greater specific root length and specific leaf area, while populations with high seed mass produced seedlings with higher initial biomass [60]. Hamerlynck et al. [61] investigated how the highly successful introduced bunchgrass, crested wheatgrass (*Agropyron desertorum* [Fisch. ex Link] J.A. Schult.), is better able to recruit seedlings and compete with downy brome relative to native perennial bunchgrasses like BBWG. They attributed these qualities to the high seed density of crested wheatgrass, double that of BBWG, which in turn has been attributed to high photosynthate production by the crested wheatgrass spike [61,89]. James et al. [32] examined survival of propagules across demographic stages during BBWG establishment in eastern Oregon, finding that most attrition of BBWG propagules occurred between germination and seedling emergence. This emergence bottleneck may relate to freezing and thawing of the seedbed, physical soil crusts, or susceptibility to pathogens.

### 4.2. Abiotic-Stress Response

Roots of BBWG have been shown to differ from those of downy brome and crested wheatgrass in their response to water. The higher root surface area of downy brome seedlings may account for their greater vigor relative to those of BBWG [74]. Compared to BBWG, downy brome seedlings are better able to avoid stress when infrequently watered, as evidenced by a less-negative xylem pressure potential, higher shoot water content, lower leaf temperature, and lower stomatal conductance [90]. Crested wheatgrass seedlings increased both shoot and root dry-matter in response to increased water availability, thereby maintaining high root mass fraction (root mass/[shoot + root mass]) [68]. In contrast, seedlings of BBWG responded to increased water by increasing shoot dry-matter while reducing root dry-matter in equal quantities, thereby lowering root mass fraction and forfeiting opportunities to capitalize on high soil water [68].

### 4.3. Competitive-Stress Response

Downy brome germinates more rapidly than BBWG, particularly under cooler temperatures [58,74,91,92]. This, coupled with opportunistic uptake of mineral nitrogen (i.e., NO_3_^−^–N and NH_4_^+^–N) during episodic high-temperature periods (>15 °C), may account for downy brome’s high nutrient-acquisition plasticity and invasiveness relative to native perennials like BBWG [36,93]. Walker et al. [94] suggested that an acquisitive strategy for nutrients may be beneficial for seedling growth, yet a nutrient conservation strategy may be paramount for longevity.

### 4.4. Defoliation-Stress Response

Bluebunch wheatgrass has long been regarded as highly susceptible to defoliation, particularly at the boot stage of phenological development [95]. This damage can be inhibited by removal of competitors [96], but it is accentuated by drought [97]. Following defoliation, both crested wheatgrass [98] and Snake River wheatgrass (*Elymus wawawaiensis* J. Carlson & Barkworth) [87,99] display superior regrowth relative to BBWG. Poor regrowth in response to defoliation is unrelated to carbohydrate reserves, as reserves in BBWG crowns were highest during the late boot stage [100], the same time when susceptibility to defoliation is greatest. In addition, regrowth was uncorrelated with carbohydrate concentration or pool size [101]. Both BBWG and crested wheatgrass seedlings show compensatory (increased) photosynthetic rate in response to defoliation [89,102]. However, in BBWG, this compensatory increase in photosynthesis reduced water use efficiency, while in crested wheatgrass, the reverse occurred. This suggests that BBWG has limited ability to cope with drying soil, especially during seedling establishment [81].

Instead of carbohydrate reserves, the superior regrowth of crested wheatgrass following defoliation relates to a relative lack of morphological constraints. Crested wheatgrass produced 17 times more daughter tillers than BBWG following defoliation [89] (Figure 8) and [101], despite similar numbers of meristematic buds [98]. In addition, crested wheatgrass allocates more resources to shoots and curtails root growth following defoliation, while BBWG roots grow unabated without apparent benefit to the plant [89]. Regrowth biomass in BBWG was correlated with tiller number but not mass per tiller [87]. Mukherjee et al. [103] reported that BBWG populations best able to compensate for defoliation were those with low undefoliated biomass, suggesting a trade-off between the above-ground productivity and defoliation tolerance.

### 4.5. Reproductive Output

Hamerlynck et al. [61] suggested that crested wheatgrass has traits that confer adaptation to a wide array of environmental conditions, enabling viable seed production in most years, while seed production of natives like BBWG is limited to years with wet spring conditions. Yield components of seed yield can be estimated through path-coefficient analysis. While no such analysis has been performed on BBWG, results are available for other perennial Triticeae species. Spikes per plant was the most important seed-yield component in Russian wildrye (*Psathyrostachys juncea* Nevski) [77], tall wheatgrass (*Thinopyrum ponticum* [Podp.] Z.-W. Liu & R.-C. Wang) [76], and western wheatgrass (*Pascopyrum smithii* Rydb.) [78], with seeds per spike and seed mass being much less important. However, all three yield components were important in basin wildrye (*Leymus cinereus* [Scribn. & Merr.] À. Löve) [79].

### 4.6. Persistence

Persistence of established BBWG plants is important because, while its persistence may be acceptable at sites with higher precipitation, it tends to be low on the drier sites that are the most difficult to establish. Ott et al. [104] evaluated results from 16-year-old 1999 seedings including Whitmar, Anatone, and Goldar BBWG at two proximal sites in west-central Utah. Cover increased over time at Mud Springs (368 mm average annual precipitation), but persistence was poor at Jericho (311 mm). Across 34 trials measured in the year following seeding and 22 trials measured three years post-seeding, Whitmar’s stand increased from 45 to 79%, while Anatone’s declined from 64 to 47% and Goldar’s declined from 63 to 37% [30]. These data corroborated the earlier findings of Hull [105], who evaluated 60 20-to-40-year-old seedings across southern Idaho. While BBWG performed ‘very poor’ overall, with many failures on the more arid sites, success was achieved with Whitmar in southwestern Idaho. Similar to Robins et al.’s [30] results, Stonecipher et al. [106] found that Goldar did not persist 12 years after seeding at Howell, UT (364 mm AAP) or Nephi, UT (375 mm). Bluebunch wheatgrass populations were similar to Siberian wheatgrass for persistence through year 5 at Cheyenne, WY (395 mm), Beaver, UT (337 mm), and Tintic, UT (372 mm) [31]. However, at Malta, ID, where precipitation is lower (270 mm), plant frequency decreased over time, with P-7 being best in year 1 and Goldar being worst in years 2–4. At year 5, BBWG stand was similar (*p* > 0.05) to that of ‘Vavilov II’ Siberian wheatgrass at Cheyenne, Beaver, and Tintic, but only 24% of Vavilov II at Malta. Collectively, these data suggest that Whitmar is more persistent than Goldar, and this difference is related to differences in long-term drought tolerance. Persistence is most problematic at drier sites, such as Malta, where stand loss is most severe and high-rainfall years are most infrequent. Thus, evaluations for persistence should take place at such sites.

## 5. Interfacing Ecological Filters and Genomics

Ecological filters represent dispersal, abiotic, biotic, or demographic processes that eliminate individuals during plant community assembly and may be controlled by management actions, e.g., seeding rates, burning, fertilization, tillage, and litter management [107]. For example, Grman et al. [108] compared the relative strength of dispersal and establishment filters for 39 species across 29 tallgrass prairie restoration plantings in southwestern Michigan by varying seeding rates and measuring soil parameters, precipitation, and plant occurrence.

Here we interface (1) an ecological trait model incorporating critical ecological traits screened by abiotic/biotic and demographic ecological filters (Figure 1, Table 1) with (2) a genomic trait model that can make phenotypic predictions based on genomic DNA sequences (genotypes). This permits direct selection for genes that control the ecological traits that facilitate filter penetration (Figure 2). However, these two models can be expected to require modification with continued selection due to changing gene frequencies. Thus, ‘retuning’ the genomic model may be required (feedback arrow on right-hand side of Figure 2). Trait assessment (middle of Figure 2) may also require retuning (feedback arrow on left-hand side of Figure 2), resulting in a modified ecological trait model.

## 6. Genomics and Selection

A genomic approach may facilitate the development of the plant materials needed to restore degraded ecosystems across a broad geographic scale [6,109]. Genomic tools have many recognized applications in biological conservation and ecological restoration that are based on the analysis of neutral and adaptive genetic variation [6]. Some of these tools have been used to elucidate genetic diversity and genetic history of BBWG populations used in restoration [51,53,110]. However, marker assisted selection (MAS; see Appendix B), and more recently genomic selection (GS), also have underutilized potential to identify favorable genetic combinations [109,111,112,113] for restoration of natural ecosystems. They can be used to construct a framework to link genome sequences with environmentally modulated plant responses [16]. Until recently, these technologies have not been utilized to develop plant materials for restoration and for enhancing ecosystem services. However, rapid proliferation of ‘next-generation’ DNA sequencing technologies over the past two decades now permits development of robust genomic resources beyond a small number of crop and livestock species [12,114,115].

### 6.1. DNA-Based Selection

The genomic technologies of MAS and GS enable the evaluation and selection of natural genetic variation at the DNA level. Traditional processes of evaluation, selection, and intermating of individuals are still required, but MAS and GS are based on relatively simple DNA markers (genotypes) instead of plant phenotypes, which are easily confounded by environmental effects. Markers may be single-nucleotide polymorphisms (SNPs) or small DNA insertions or deletions (indels) that usually have only two alternative forms. These genomic tools enable scientists to overcome inherent challenges to the adequate evaluation of complex traits, i.e., those subject to a mixture of complex genetic and confounding non-genetic effects in genetically heterogeneous populations.

In the past, extensive replication of field studies across a variety of locations has been required to accurately characterize complex traits [116]. For self-pollinating species, such as many cereal crops, requirements for experimental replication can be satisfied by using seed of inbred lines. However, most perennial plant species, including BBWG, are allogamous (cross-pollinating) and self-incompatible, resulting in genetically diverse individuals. In these cases, replication can be facilitated by clonal propagation, but a more efficient statistical alternative is to determine breeding values using best linear unbiased prediction (BLUP). A BLUP value is based not only on the performance of the individual, but also on the performance of its relatives weighted by their genetic relationship, as approximated by pedigree information [117,118,119]. Genomic technologies provide the exact genetic relatedness among individuals, rather than the expected value based on pedigree.

### 6.2. Genomic Selection

The main goal of GS is to develop models to accurately predict breeding values, i.e., genome-estimated breeding values (GEBVs), of individuals using DNA-sequence or DNA-marker information. In contrast to MAS (see Appendix B), the more generalized GS approach utilizes many DNA markers with sufficient marker density to detect and select favorable DNA linkages. The accuracy of GEBVs is a function of the number of markers and the size of the training population, i.e., the number of individuals with known genotypes and phenotypes based on training records. Because the number of markers typically exceeds the size of the training population, thereby creating an estimation problem, special statistical methods are utilized to address this issue (see Appendix C).

Notwithstanding technical and mathematical complexities, the potential acceleration of breeding efficiencies enabled by GS is impressive [12,118,120,121]. The rapid proliferation of genomic resources opens new avenues for application of MAS and GS for restoration of native plant species threatened by introduced pathogens or native pathogens that are increasing with climate change [122]. For example, genetic linkage maps and a draft genome reference sequence for the American chestnut will facilitate the application of MAS and GS to select for resistance to an alien blight fungus (*Cryophonectria parasitica*) in this keystone tree species [109,112,123,124].

Although breeding has the potential to facilitate restoration of wild plants and animals, any potential loss of genetic diversity due to breeding is a concern [125,126], especially for rare or endangered species where genetic diversity has already been restricted [127,128]. The loss of genetic diversity is always a concern in breeding because the gains from selection are closely related to the amount of heritable genetic variation [129,130]. Compared to pedigree-based phenotypic selection, genomic selection may result in a more rapid decline in genetic variation and selection response [131,132]. However, it has also been shown that genomic selection can maintain higher levels of diversity because it measures genetic relatedness more accurately [129,133,134,135]. One particular problem with traditional phenotypic selection in self-incompatible, wind-pollinated grasses is that the pollen parents are usually unknown and the success of pollen parents is non-random [115]. Paternity analysis using genomic DNA markers has shown that some pollen parents are much more successful than others, which reduces the effective population size if genomic relationships are not controlled in the selection process [115]. Methods of genomic selection are now being developed to both maximize genetic gains and maintain the high levels of genetic variation needed to maintain future selection response [133,136,137]. These methods have potential applications for improving restoration outcomes.

## 7. Conclusions

We believe an ecological genomics model, coupled with quantitative assessment of plant functional traits, will provide a new essential toolkit for arid land restoration efforts. The most powerful aspect of this approach is that it may predict plant performance across the full range of testable demographic processes. Such efforts are critical in face of the ongoing increase in regional climate variability associated with global climate change, which will likely accelerate rangeland degradation. Indeed, Kilkenny’s [138] climate model has predicted that BBWG is vulnerable to extirpation in the Snake River Plain and Columbia Plateau due to warming temperatures. Whitham [3] lamented the lack of good plant-material options for populations compromised at the warm end of species’ distributions. Furthermore, seed-transfer options may be limited due to the emergence of no-analog ecosystems as a response to climate change [139,140,141,142]. A justification for developing more effective native plant materials can be made on this basis. While in theory, natural selection will eventually produce well-adapted native species, existing and pending ecological and economic consequences require more immediate action to address the huge areal extent of this problem.

## Figures and Tables

**Figure 1 ijms-23-08275-f001:**
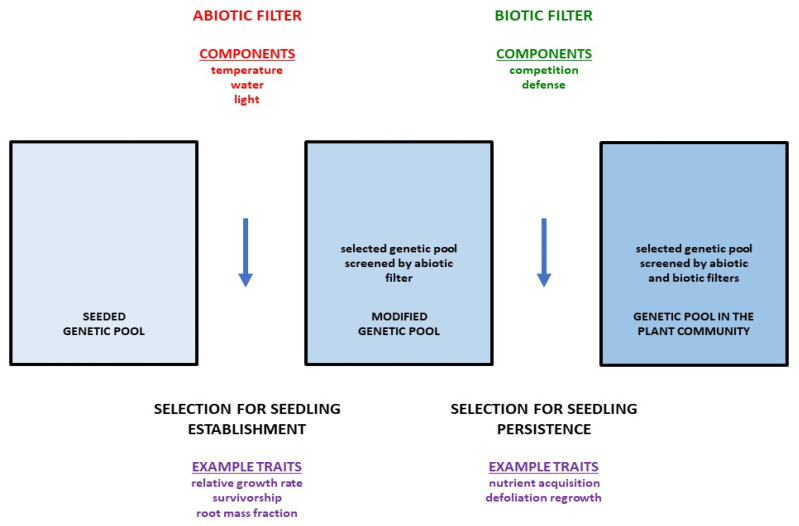
Sorting of undesirable from desirable phenotypes in the seeded genetic pool through selection for seedling establishment (modulated primarily by a biotic filter) and seedling persistence (modulated primarily by an abiotic filter), ultimately resulting in the genetic pool present in the plant community.

**Figure 2 ijms-23-08275-f002:**
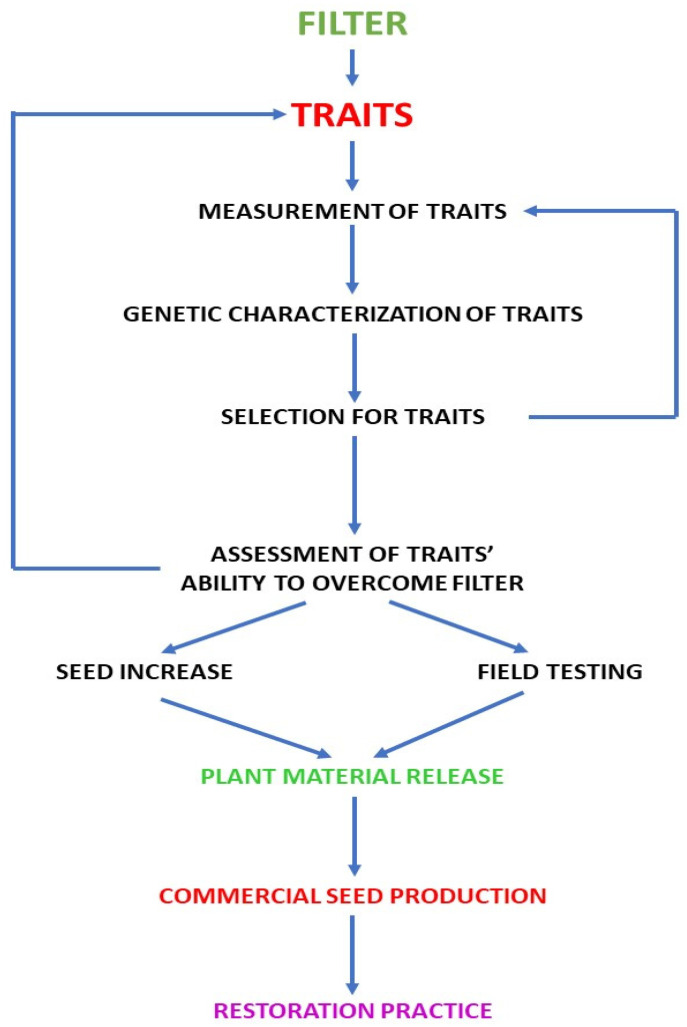
A flowchart for development of plant materials and their subsequent delivery to the marketplace.

**Table 1 ijms-23-08275-t001:** Demographic filters and associated traits.

Demographic Filters/Traits	Trait Category ^1^	Units	References
*Germination*			
rate at room temperature	1	index value	[56,57]
rate at 10 °C	1, 2	index value	[56,58]
seed mass	1	mg seed^−1^	[59]
*Emergence*			
seed mass	1, 2	mg seed^−1^	[57,60]
seed density	1, 2	mg cc^−1^	[61]
deep-seeding (5 cm) emergence	2	%	[62]
emergence rate (10 °C)	1, 2	index value	[56,63]
coleoptile density	2	dry weight per unit fresh weight	[59]
elongated subcoleoptile internode	2	frequency (%), length (mm)	[64]
*Seedling establishment*			
cold-temperature (10 °C) growth	1, 2, 3	mg seedling^−1^	[60]
root biomass	1, 2, 3	mg seedling^−1^	[65]
root:shoot biomass	1, 2, 3	mg mg^−1^	[66,67,68]
root length	2, 3	cm seedling^−1^	[58]
specific root length	2, 3	cm g^−1^	[69,70]
root tips	3	number seedling^−1^	[71]
tiller recruitment (number and stage)	1, 4	tillers seedling^−1^ stage^−1^	[72]
specific leaf area	1, 2, 3	cm^2^ g^−1^	[73,74]
recovery from 70% defoliation	4	mg regrowth seedling^−1^	[75]
*Reproductive output*			
spike number	5	spikes plant^−1^	[76,77,78]
seeds per spike	5	seeds spike^−1^	[79]
seed mass		mg seed^−1^	[73,80]
*Persistence*			
change in stand	6	% (year 1 vs. year 3, year 5)	[30]
change in spike number	6	spikes plant^−1^ (year 2 vs. year 5)	
*Developmental physiology*			
shoot biomass	1	G	[75,81]
root biomass	1, 2	G	[75,81]
root mass fraction	1, 2, 3	g g^−1^	[68]
height	1, 3	cm	[73]
dark respiration	1	mmol mol m^−2^ s^−1^	[82]
net assimilation rate	1	mmol mol m^−2^ s^−1^	[61,81,83]
carboxylation efficiency	1	mmol mol m^−2^ s^−1^	[61]
instantaneous water-use efficiency	2, 3	net assimilation rate per unit of stomatal conductance	[81]
water-use efficiency (integrated over time) measured by carbon-isotope discrimination	2	δ ^13^C/^12^C (%)	[84,85,86]
recovery from 10-cm defoliation	4	spikes plant^−1^	[87]

^1^ Trait categories are (1) growth, (2) abiotic-stress response, (3) competitive-stress response, (4) defoliation-stress response, (5) reproductive output, and (6) persistence.

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
