# Peer review of "Using Genomic Selection to Develop Performance-Based Restoration Plant Materials"

_ijms, 2022, doi:10.3390/ijms23158275_

Round 1
Reviewer 1 Report
In this manuscript, T.A. Jones described a breeding approach or pipeline based on genomic selection to develop plant materials for land rehabilitation, which is a key step towards a sustainable ecosystem. Authors addressed that native plant materials can be more efficiently and accurately selected by genomic technologies, like next generation DNA sequencing and GWAS. A detailed roadmap was raised in the manuscript for researchers to follow in the community. The concept was carefully brought in an organized and logic manner and the manuscript was well written. I only have minor issues regarding paper structure and formation.
1. In figure 2, the shape and size of the letters seems to be “shrinked”.
2. Authors should stringently follow publish guidelines of IJMS to make sure all fonts in the text and figures meet the criteria.
3. Some bold letters were noticed in the text (e.g. workhorse and Germplasm in line 235-237). Bold letters were normally not allowed in the text unless specific reasons were given in the manuscript.
4. Please stringently follow the nomenclature for publication. For example, the title of listed cited publications should be “References”, not “Literature cited”.
5. In the funding part, specific funding reference numbers and the awardees should be disclosed.
6. The required “supplemental materials” section is missing. Authors should consider add this section, which is very important for original data presentation and experimental recapitulation. Especially a lot of math models (like regression and LASSO) were mentioned. Authors should present the specific parameters they used in these models, at least, in the supplemental materials.
Author Response
Authors' responses are in caps:
- In figure 2, the shape and size of the letters seems to be “shrinked”. THIS FIGURE HAS BEEN REPLACED WITH ONE OF BETTER QUALITY.
- Authors should stringently follow publish guidelines of IJMS to make sure all fonts in the text and figures meet the criteria. THE TEMPLATE WAS USED TO ENSURE CONFORMITY TO IJMS STANDARDS.
- Some bold letters were noticed in the text (e.g. workhorse and Germplasm in line 235-237). Bold letters were normally not allowed in the text unless specific reasons were given in the manuscript. BOLDING WAS REMOVED, AS WELL AS THE GLOSSARY TO WHICH THEY REFERRED.
- Please stringently follow the nomenclature for publication. For example, the title of listed cited publications should be “References”, not “Literature cited”. DONE.
- In the funding part, specific funding reference numbers and the awardees should be disclosed. NO EXTRAMURAL FUNDING FOR THIS PAPER.
- The required “supplemental materials” section is missing. Authors should consider add this section, which is very important for original data presentation and experimental recapitulation. Especially a lot of math models (like regression and LASSO) were mentioned. Authors should present the specific parameters they used in these models, at least, in the supplemental materials. NO 'SUPPLEMENTAL MATERIALS' SECTION IS NECESSARY, AS THIS IS A REVIEW ARTICLE RATHER THAN A RESEARCH ARTICLE. THE MODELS WERE PROPOSED RATHER THAN ACTUALLY USED.
Reviewer 2 Report
The peer-reviewed article « Using genomic selection to develop performance-based restoration plant materials» is appropriate for the IJMS journal's subject and the corresponding special issue. However, the manuscript is not of proper quality. The text of manuscript is not presented according to the journal's rules. There are no sections, their numbering is according to the rules of the journal. The reference sheet is also presented incorrectly. I strongly encourage to read instruction for authors (https://www.mdpi.com/journal/ijms/instructions). I have the impression that the team either mixed up the choice of the journal, or specifically did not bother to read the instructions for the authors.
Other questions and comments:
1. Why do authors give a glossary instead of an introduction? this is completely incomprehensible to me. I consider it redundant. I think that the researchers interested in plant breeding and genomic selection are well acquainted with terminology.
2. For what purpose the authors added box 1. The relevance of the restoration of sagebrush-steppe plant communities? A photograph of unacceptable quality for publication in a highly impacted journal. What does the photo show??? And what do the authors mean by box?
3. pp. 15-16, 30-32 and 35-36. Why is it tinted yellow? Is it for beauty or does it have some role?
4. The manuscript is missing boxs 2 and 3.
Summary
The manuscript is presented in poor quality and is not made according to the rules of the journal. The article should be rejected rewritten and resubmetted.
Author Response
THE AUTHORS' RESPONSES ARE IN FULL CAPS.
The peer-reviewed article « Using genomic selection to develop performance-based restoration plant materials» is appropriate for the IJMS journal's subject and the corresponding special issue. However, the manuscript is not of proper quality. The text of manuscript is not presented according to the journal's rules. There are no sections, their numbering is according to the rules of the journal. The reference sheet is also presented incorrectly. I strongly encourage to read instruction for authors (https://www.mdpi.com/journal/ijms/instructions). I have the impression that the team either mixed up the choice of the journal, or specifically did not bother to read the instructions for the authors. THIS TIME WE USED THE TEMPLATE PROVIDED.
Other questions and comments:
- Why do authors give a glossary instead of an introduction? this is completely incomprehensible to me. I consider it redundant. I think that the researchers interested in plant breeding and genomic selection are well acquainted with terminology. THE GLOSSARY HAS BEEN DROPPED.
- For what purpose the authors added box 1. The relevance of the restoration of sagebrush-steppe plant communities? THE PURPOSE OF BOX 1 (NOW APPENDIX 1) IS TO PROVIDE CONTEXT FOR THE APPROACH. A photograph of unacceptable quality for publication in a highly impacted journal. What does the photo show??? THE PHOTO HAS BEEN DROPPED. And what do the authors mean by box? BOXES ARE NOW APPENDICES.
- pp. 15-16, 30-32 and 35-36. Why is it tinted yellow? Is it for beauty or does it have some role? COLORATION HAS VED. REMO
- The manuscript is missing boxs 2 and 3. THESE ARE NOW APPENDICES 2 AND 3.
Summary
The manuscript is presented in poor quality and is not made according to the rules of the journal. The article should be rejected rewritten and resubmetted.
USING THE TEMPLATE, THIS REVISION NOW CONFORMS TO IJMS FORMATTING.
Round 2
Reviewer 2 Report
The authors used the template and rules to submit to the IJMS journal. The resubmitted manuscript takes into account the comments.